# Enhanced Vascular Bifurcations Mapping: Refining Fundus Image Registration

Jesús Eduardo Ochoa-Astorga [1], Linni Wang [2], Weiwei Du [1,*] and Yahui Peng [3]

1  Information and Human Science, Kyoto Institute of Technology University, Kyoto 6068585, Japan; d1821501@edu.kit.ac.jp
2  Retina & Neuron-Ophthalmology, Tianjin Medical University Eye Hospital, Tianjin 300084, China; wln5156@163.com
3  School of Electronic and Information Engineering, Beijing Jiaotong University, Beijing 100044, China; yhpeng@bjtu.edu.cn
*  Correspondence: duweiwei@kit.ac.jp; Tel.: +81-075-724-7445

**Abstract:** Fundus image registration plays a crucial role in the clinical evaluation of ocular diseases, such as diabetic retinopathy and macular degeneration, necessitating meticulous monitoring. The alignment of multiple fundus images enables the longitudinal analysis of patient progression, widening the visual scope, or augmenting resolution for detailed examinations. Currently, prevalent methodologies rely on feature-based approaches for fundus registration. However, certain methods exhibit high feature point density, posing challenges in matching due to point similarity. This study introduces a novel fundus image registration technique integrating U-Net for the extraction of feature points employing Fundus Image Vessel Segmentation (FIVES) dataset for its training and evaluation, a novel and large dataset for blood vessels segmentation, prioritizing point distribution over abundance. Subsequently, the method employs medial axis transform and pattern detection to obtain feature points characterized by the Fast Retina Keypoint (FREAK) descriptor, facilitating matching for transformation matrix computation. Assessment of the vessel segmentation achieves 0.7559 for Intersection Over Union (IoU), while evaluation on the Fundus Image Registration Dataset (FIRE) demonstrates the method's comparative performance against existing methods, yielding a registration error of 0.596 for area under the curve, refining similar earlier methods and suggesting promising performance comparable to prior methodologies.

**Keywords:** fundus image registration; feature extraction; blood vessels segmentation; feature matching; enhanced vascular bifurcations mapping

## 1. Introduction

The retina is a crucial part of the human visual system, converting optical stimuli into neuroelectric signals processed by the brain. Fundus imaging is the primary method for assessing the retina, playing a key role in diagnosing various ocular pathologies like diabetic retinopathy, age-related macular degeneration, glaucoma, among others [1]. Integrating multiple fundus images and registering them enhances this process significantly. It complements individual assessments, offering a more comprehensive evaluation that assists physicians in diagnosing retinal diseases more effectively [2].

Fundus image registration involves aligning images with overlapping regions by establishing correspondences between them. This feature-based registration is crucial for various applications in retinal analysis [3,4], such as longitudinal studies, which examine morphological changes in ocular structures over time by comparing fundus images taken at different intervals. Another important application is image mosaicking, which combines multiple fundus images from different viewpoints to broaden the visual perspective. However, analyzing multiple perspectives is challenging for healthcare practitioners due to the limited field of view of fundus images [5], typically around 45°. Although wide-angle

fundus photography offers a solution with a potential field of view exceeding 100° [6], its adoption is restricted due to requirements like pupil dilation and the high cost of specialized cameras, especially disadvantaging patients in rural areas [7]. Similarly, efforts to improve resolution through the super-resolution of fundus images rely on registration techniques to align low-resolution images from portable fundus cameras often used in telemedicine initiatives [8]. By synthesizing these slightly different images, resolution enhancement occurs, improving visual acuity and revealing clinically significant details that were previously obscured.

Several techniques have been developed for registering fundus images, with some focusing on initial blood vessel detection. For example, the Straightforward Bifurcation Pattern-Based Fundus Image Registration method (SBP-FIR) described in [9] relies on pixel-wise segmentation, while the approach in [10] utilizes the Frangi filter [11] for detecting tubular structures in fundus images. These methods employ thresholding-based segmentation and feature-based segmentation using filtering techniques, respectively. While these approaches offer advantages in terms of ease of implementation, simplicity, and improved noise robustness for filtering-based methods, this study employs deep learning-based segmentation for its higher accuracy and adaptability to diverse datasets, reducing dependency on threshold selection.

This paper introduces an algorithm for registering pairs of fundus images. Unlike previous methodologies that relied on thresholding-based segmentation and the Frangi filter [9–11] for blood vessel delineation, this approach utilizes U-Net [12] to identify the specific region of interest within the image. The primary advantage of using U-Net over the Frangi filter lies in U-Net's autonomy in sensitivity parameter settings and its more efficient execution time for the entire segmentation process. Configuring sensitivity parameters for the Frangi filter limits its adaptability and complex structure detection, while U-Net segmentation methods proficiently learn intricate features without such constraints. Additionally, incorporating U-Net ensures accurate identification of bifurcation regions, a feature occasionally lacking in the Frangi filter application [13]. The process begins with U-Net identifying the blood vessels' region of interest, followed by thinning, pattern detection, and characterization of feature points using the Fast Retina Keypoint (FREAK) descriptor [14]. Feature matching, outlier removal, and computation of the similarity transformation are accomplished through Random Sample Consensus (RANSAC) [15]. Finally, a seamless image is generated through blending. This method aims to enhance registration precision and reduce execution time, crucial factors for introducing new methodologies into clinical practice for computer-aided diagnosis. Evaluation on both segmentation results and registration accuracy yields an Intersection over Union (IoU) score of 0.7559 on the Fundus Image Vessel Segmentation (FIVES) dataset [16] and an Area Under the Curve (AUC) of 0.596 on the Fundus Image Registration Dataset (FIRE) dataset [17]. It competes with complex methods and notably reduces execution time by half compared to one of the top methods in registration accuracy while maintaining competitive performance in certain categories.

This paper's main contribution is the novel integration of U-Net and FREAK descriptor for fundus image registration, aiming to refine existing methods by employing the FIVES dataset as a new dataset for segmentation of blood vessels in fundus images. This method initiates region-of-interest identification through U-Net. Subsequent feature extraction results in a more evenly distributed layout of feature points across fundus images compared to other state-of-the-art extractors. Although it may exhibit slightly lower accuracy in certain aspects compared to other methodologies, this technique significantly reduces execution time. Evaluation across diverse datasets, including the FIVES dataset [16] and the FIRE dataset [17], highlights its competitive performance against complex methods while significantly reducing computational load.

The paper is structured as follows: Section 2 offers a comprehensive review of relevant studies within the field. Section 3 outlines the proposed approach for fundus image registration. Section 4 provides a detailed examination of the experimental outcomes

resulting from the application of the proposed technique. Following this, Section 5 presents the discussion. Lastly, Section 6 concludes the paper.

## 2. Related Work

In the extensive literature on fundus imaging, numerous methods have been developed for registering and stitching fundus images. While the Scale-Invariant Feature Transform (SIFT) [18] is widely regarded as a robust and commonly used technique in image processing, its application to identifying corresponding points in fundus images faces challenges. These challenges are particularly evident in scenarios involving fluctuating illumination or when dealing with surfaces that exhibit similar intensity levels. This difficulty arises because SIFT relies solely on gray information for feature extraction, making it challenging to distinguish between visually analogous conditions.

However, some studies have utilized the SIFT descriptor, such as in [19], to identify feature points corresponding to bifurcations in fundus images. Following feature matching and false match removal, a Voronoi diagram is used to create mosaic images. Yet, these methods are typically assessed only for this task, overlooking other applications like longitudinal studies and super-resolution imaging. In our study, we assessed these techniques on the Fundus Image Registration Dataset (FIRE) [17], also employed in [20], a leading study on this dataset. The approach in [20] integrates blood vessel bifurcations and the SIFT detector as feature points, initially estimating the camera pose through RANSAC using a spherical eye model for precise results. Subsequent steps involve parameter estimation for an ellipsoidal eye model and further camera pose refinement. While this methodology is precise, employing two feature detectors leads to an increased number of detected feature points. Consequently, matching and registration become computationally complex, resulting in slower processing speeds. On the contrary, Gong et al. in [21] argue that intensity-based registration outperforms methods reliant on feature-point registration. In this approach, dimensionality reduction is crucial, transforming input images into a lower-dimensional space for efficient recognition of global correspondences. Specifically, in the context of contiguous image pairs, their method optimizes the Mutual Information metric for registration before synthesizing panoramas through image blending.

Regarding works that have completely excluded the use of SIFT, numerous feature-based registration methodologies relying solely on landmark identification along blood vessels have been proposed. In the study referenced as [22], a Convolutional Neural Network (CNN) trained on the Digital Retinal Images for Vessel Extraction (DRIVE) dataset detects vascular crossovers and bifurcations using the U-Net architecture. This specific U-Net architecture predicts a heatmap identifying landmarks. Similarly, the use of the Deep Retinal Image Understanding network (DRIU) alongside pre-trained VGG-16, as described in [23], involves a preliminary stage of blood vessel segmentation to facilitate subsequent feature detection. Furthermore, alternative methodologies not utilizing deep learning have focused on blood vessels as a focal area of interest for finding feature points suitable to fundus registration. For example, the study detailed in [24] conducted blood vessel segmentation through fundamental morphological operations and curvature evaluation. Likewise, in the work explained by [25], the vessel tree served as a focal region for feature extraction, finding bifurcations extracted from the segmented blood vessel and incorporating a Bayesian approach as a matching algorithm. A straightforward mosaicking of fundus images methodology is presented in [26], in which a CNN model segments vascular structures within fundus images, specifically targeting the detection of vascular bifurcations. These bifurcations are subsequently extracted as feature points upon the vascular mask, and the estimation of transformation parameters for image stitching is established among these vascular bifurcations. While the effectiveness of this method was evaluated across a limited set of eyes, the primary limitation of this study lies in its absence of comparison with other methods to support its performance.

The investigation into bifurcations and crossovers in fundus images is crucial for accurately identifying blood vessel issues and locating clots, essential for precise medical

intervention [27]. These junction points play a vital role in understanding variations in blood flow and pressure dynamics within the vessels [28]. In [27], a meticulous approach involved segmenting patches of 21 × 21 pixels along vessel structures obtained from binary segmentations. These patches were used as training data for a Res18 convolutional neural network to distinguish these features. Despite a modest dataset of 40 images from the DRIVE database (30 for training, 10 for testing), the patch-based method generated an extensive dataset of over 100,000 patches, ensuring a robust learning process. Similar techniques were employed in other studies, such as Patwari et al. [29] and others [30,31], utilizing methods like morphological skeletonization, image enhancement through histogram equalization, and the extraction of bifurcation points from blood vessel skeletons, thereby enhancing the understanding of vessel structures and their features.

Several methods in fundus image analysis focus on blood vessel segmentation to detect bifurcation points. This paper's method emphasizes segmenting blood vessels to identify feature points within the segmentation map. Numerous studies concentrate on employing Convolutional Neural Networks (CNNs) for this purpose. For instance, Dharmawan et al. [32] proposed a modified U-Net [12] for patch-based segmentation, reducing downsampling operations and incorporating dropout layers between consecutive convolutional layers. They trained this network using cross-entropy loss on datasets such as DRIVE [33], STARE [34], and HRF [35]. In contrast, Ref. [36] introduces ResWnet, modifying the U-Net structure by minimizing downsampling layers to two and implementing an encoding–decoding–encoding–decoding structure. ResWnet enhances feature retention and semantic extraction by utilizing skip connections and residual blocks, improving sensitivity across various vessel scales, as evaluated on DRIVE and STARE databases. Moreover, Ref. [37] presents DRNet, a method inspired by U-Net and utilizing a deep dense residual network structure. DRNet merges feature maps across blocks, aiding spatial reconstruction, and introduces DropBlock to address overfitting issues. Each method showcases innovations in vessel segmentation approaches for fundus image analysis.

Differing from some previous research approaches, this study primarily focuses on aligning fundus images through a sequential process. Initially, it employs the traditional U-Net architecture [12] for segmenting blood vessels, utilizing the Fundus Image Vessel Segmentation (FIVES) dataset [16]. This unique dataset contains a significant collection of annotated fundus images, reportedly offering superior accuracy in labeling compared to other publicly available datasets designed for blood vessel segmentation in fundus images. The detection of bifurcations occurs along the blood vessel skeleton to establish the geometric relationship between the images. Subsequently, an image blending technique is applied to generate the final aligned result.

## 3. Proposed Method

This section introduces the conceptual framework of the proposed method, which aims to develop a feature-based fundus image registration technique. The key objective of this paper is to leverage bifurcations and crossovers along the morphological skeleton of blood vessel segmentation as feature points. The SBP-FIR method outlined in [9] demonstrated promising performance in registering pairs of fundus images. This proposal initially employs a pixel-wise segmentation method to define a region of interest over blood vessels, commonly used for detecting bifurcations and crossovers in fundus images [27–31]. After skeletonization, a pattern detection process is utilized to locate bifurcations and crossovers, followed by their characterization using Histogram of Oriented Gradients (HOG). This characterization facilitates establishing a geometric relationship between the fundus images, ultimately enabling image warping and blending for registration purposes. The proposed method unfolds across four principal stages: feature extraction, feature matching, computation of the transformation matrix and subsequent image warping, concluding with image blending, as illustrated in Figure 1. Unlike the approach in [9], this method utilizes a deep learning-based technique for blood vessel seg-

mentation. The primary objective is to enhance segmentation accuracy while concurrently reducing processing time, complemented by the utilization of the FREAK descriptor.

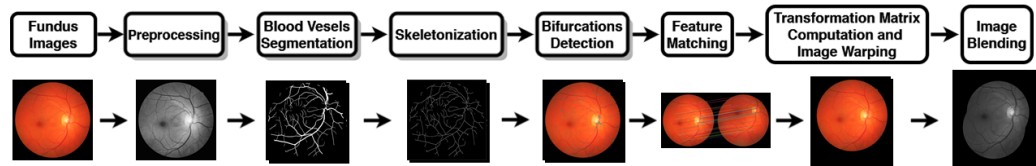

**Figure 1.** The flowchart illustrating the proposed method delineates sequential stages for processing both source and target images. Each step is accompanied by paired images, with stages shadowed in the background indicating steps applied individually to images.

### 3.1. Flowchart Description

Figure 1 presents an outline of the proposed method. Initially, a pair of color fundus images constitutes the input. Feature point extraction prioritizes the blood vessels area, followed by blood vessels segmentation on the original fundus images. This is succeeded by thinning or skeletonization to detect bifurcation patterns in the thinned images. Subsequently, descriptors represent feature points in each image, facilitating their matching for identification of shared points between the source and target images. Once the relationship between points in the source and target images is established, a transformation matrix is computed and applied to warp the source image. Finally, a blending process ensures seamlessness in the resulting image, mitigating visible seams due to potential exposure variations in the overlapped regions.

The proposed method demonstrates significant advancements compared to its predecessor [9], primarily attributed to a shift in the segmentation approach and feature extraction techniques. Transitioning from a pixel-wise segmentation to leveraging U-Net for blood vessel segmentation markedly heightens accuracy [38] and accelerates processing times. This adoption of deep learning enhances segmentation precision and optimizes computational demands. Additionally, employing the FREAK descriptor in feature extraction replaces the Histogram of Oriented Gradients (HOG) descriptor, aiming to accelerate the process without compromising accuracy [39]. Moreover, refining the matching process by omitting certain verification methods from the previous approach increases efficiency, allowing for faster registration while preserving potential matches. These strategic modifications collectively enhance the speed and precision of image registration, positioning the new method as a significant improvement over its predecessor. Further elaboration on the method is provided in subsequent sections.

### 3.2. Feature Extraction

Methods in fundus image registration that utilize the entire image for alignment and determine error based on similarity measures often struggle when aligning the complete fundus image surface [40]. In the proposed method for fundus image registration, feature point detection relies on vessel extraction from the fundus image. Hence, a robust method for detecting the vascular structure is necessary.

For this, the preprocessing stage begins with extracting the green channel of the fundus image, as it typically exhibits greater contrast between blood vessels and background. However, fundus images often appear dark, so to enhance brightness and contrast, gamma correction is applied first, as described by Equation (1):

$$y = 255 \left( \frac{x}{255} \right)^{\frac{1}{\gamma}} \tag{1}$$

where $x$ is the original pixel, $y$ is the resultant pixel and $\gamma$ is the gamma correction value. For the proposed method, $\gamma$ was set to $\gamma = 1.5$ [41]. Through this process, the tonal response of the image is adjusted, emphasizing the darker areas of the fundus image while compressing the brighter ones. Subsequently, preprocessing continues with the application

of Contrast Limited Adaptive Histogram Equalization (CLAHE) [42], enhancing the visual quality of the fundus image while avoiding artifacts that other traditional histogram equalization methods may produce. The preprocessing steps are illustrated in Figure 2.

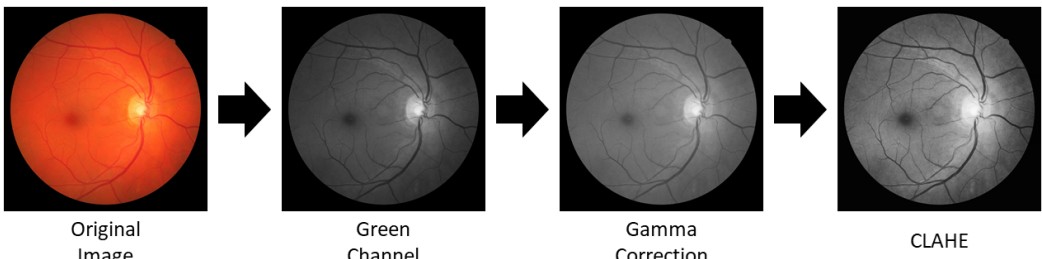

| Original Image | Green Channel | Gamma Correction | CLAHE |

**Figure 2.** Preprocessing steps preceding blood vessel segmentation within the fundus image.

Following preprocessing, the subsequent stage in feature extraction for the fundus image involves blood vessel segmentation. For this task, patch-based segmentation is conducted using U-Net [12], as employing the entire image for both training and prediction strategies hinders U-Net from achieving satisfactory results in vessel segmentation [43]. Traditional segmentation methods are avoided in the proposed method due to evidence indicating that deep learning-based approaches are often faster and more accurate [44], showcasing superior performance compared to human experts in retinal vessel segmentation [38].

Processing an entire high-resolution fundus image in one go can require significant memory resources, especially for deep learning models. Therefore, to reduce the memory usage demanded during training or inference, patch-based segmentation is conducted. This type of segmentation involves three basic steps: partitioning (dividing the fundus image into smaller patches), processing (performing blood vessel segmentation), and aggregation (combining the segmented patches to generate a final segmented output for the entire fundus image). It offers several benefits, such as the ability to handle large images, enhance local context capture, robustness to variations, data augmentation, and generalization, among others.

The proposed methodology utilized the classical U-Net architecture, structured with convolutional blocks employing a downsampling schema facilitated by max pooling. Each block integrates $3 \times 3$ convolutional layers activated by Rectified Linear Units (ReLU) to ensure feature extraction and representation. The upsampling component restores the spatial resolution while preserving vital skip connections, culminating in an output layer comprising $1 \times 1$ convolutional layers with sigmoid activation. Designed to the task of blood vessels segmentation within fundus images, this approach performs a patch-based segmentation strategy using U-Net. The focus is on bifurcation detection, so the network undergoes exclusive training on patches containing at least one bifurcation, guiding the model to recognize these essential points. By utilizing the FIVES dataset [16]—a repository containing 800 high-resolution color fundus images, meticulously annotated at the pixel level—the network navigates its learning process.

The training consists of distinct phases aimed at refining the model's understanding. Initially, 2000 patches are utilized, with a learning rate of $1 \times 10^{-4}$ over 20 epochs. This initial training is followed by two fine-tuning stages, involving entirely new patch imagery. The first fine-tuning phase utilizes 2000 new patches, extending through 20 epochs with a reduced learning rate of $3 \times 10^{-5}$. Subsequently, the second fine-tuning phase intensifies the learning with 3000 patches, reduced to 5 epochs while refining towards a learning rate of $1 \times 10^{-5}$. This detailed approach aims to enhance the U-Net model's ability to delineate blood vessels and increase its sensitivity to identifying crucial bifurcation points within fundus images, leveraging a sequenced training methodology for optimal performance. Figure 3 illustrates the segmentation result for patches and for the entire image after the aggregation step, producing the blood vessel segmentation mask.

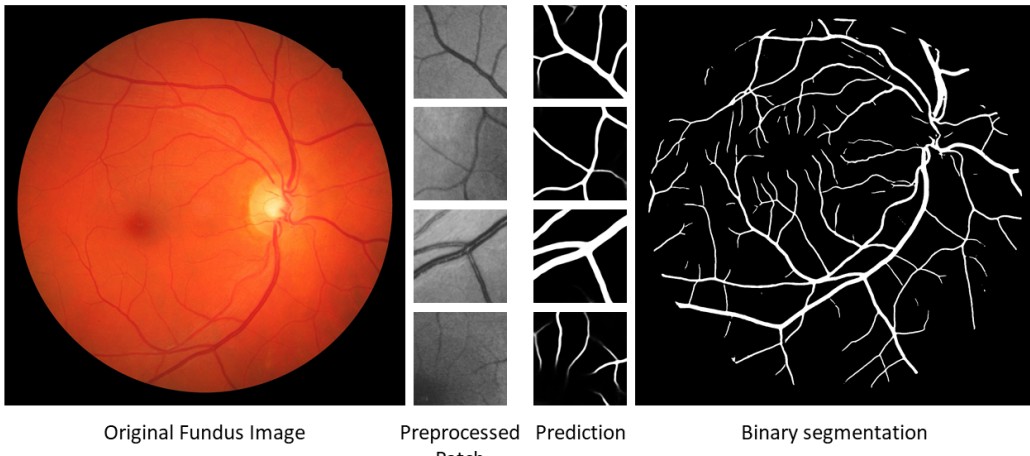

| Original Fundus Image | Preprocessed Patch | Prediction | Binary segmentation |

**Figure 3.** (On the left side) Depicted are the original fundus images and the patches extracted after preprocessing. (On the right side) Presented are the predictions of the preprocessed patches and the final fundus image segmentation with composite result derived from integrating all the segmented patches.

Next, the Zhang-Suen thinning algorithm [45] is employed to perform thinning on the image, with the aim of extracting the center lines of blood vessels previously segmented by the U-Net model. This specific algorithm works by isolating the central pathways within a binary image, achieved through the elimination of image contour points and retaining only those points that constitute the skeleton structure, thus preserving the bifurcation points within the skeleton of the fundus image blood vessels. Through multiple iterations, the algorithm progressively refines and consolidates these skeletons to derive the final representation of vascular centerlines. The outcome of applying this algorithm to the blood vessel segmentation of a fundus image is visually depicted in Figure 4.

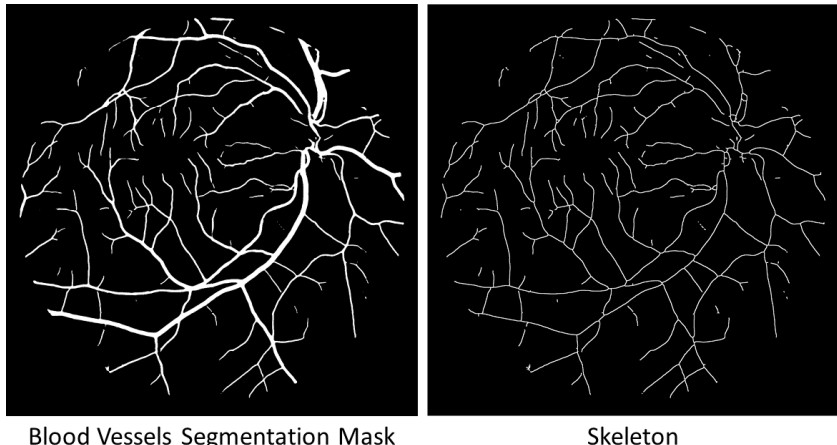

| Blood Vessels Segmentation Mask | Skeleton |

**Figure 4.** Segmentation mask illustrating blood vessels within a fundus image alongside the outcome post-application of the thinning algorithm to derive the vessel segmentation skeleton.

Finally, feature points are identified by analyzing patterns within the skeleton. Different orientations of T- and Y-shaped patterns are established to detect these specific configurations of foreground and background pixels within the vessel segmentation skeleton. The Hit-or-Miss transform, a binary morphological operation using two structuring elements ($B_1$ and $B_2$), is employed to represent both the foreground (i.e., the morphological skeleton of blood vessel segmentation) and the background of the searched patterns. Equation (2) defines the bifurcation patterns observed across the skeleton image ($A$):

$$A \circledast B = (A \ominus B_1) \cap (A^c \ominus B_2) \tag{2}$$

Here, symbols ⊛, ⊖, and ∩ denote convolution, erosion, and intersection operators, respectively. *B* combines structuring elements, where $B_1$ signifies the pattern's skeleton and $B_2$ signifies its background. Additionally, $A^c$ denotes the complement of the skeleton image.

In this context, bifurcations are identified by finding vessel pixels in their 8-neighborhood that have three non-adjacent vessel pixels. These particular patterns are illustrated in Figure 5.

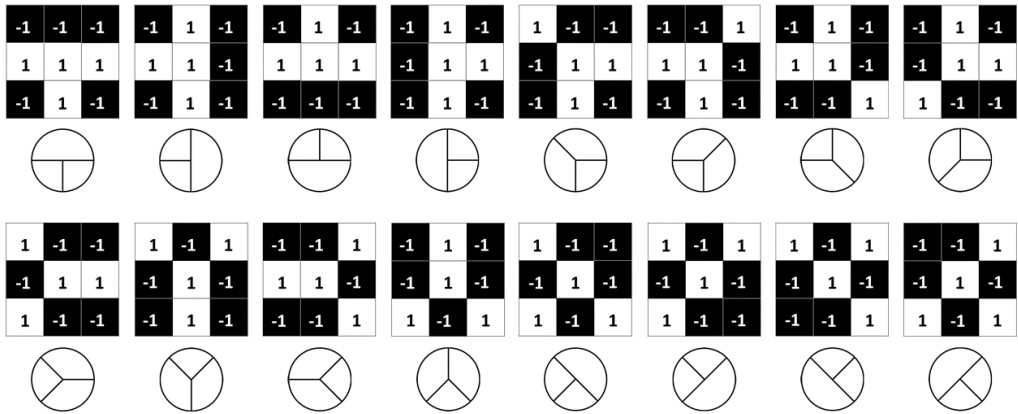

**Figure 5.** The structuring elements (**top**) illustrate the patterns (**bottom**) used to detect bifurcations within the skeleton of fundus images, where in the structuring elements, 1 represents the foreground (i.e., blood vessel skeleton) and −1 represents the background.

### 3.3. Feature Matching

Once the feature points within the fundus image are identified, establishing the geometric relationship between images involves associating features from the source image with their counterparts in the target image. Initially, bifurcation points undergo characterization via a feature descriptor to identify their corresponding points in the other image. Given the limited texture in fundus images, primarily dominated by the background, a robust feature descriptor becomes crucial. For this purpose, the FREAK descriptor is employed, known for its computational and storage efficiency, as well as its effectiveness in matching through Hamming distance [46], ensuring reliable accuracy and robustness.

The FREAK descriptor is inspired by the human retina, with the goal of mimicking retinal photoreceptors using pixels. To accomplish this, the descriptor utilizes a configuration of partially overlapped receptive fields arranged in seven rings, each containing six receptive fields. This arrangement simplifies the construction of the descriptor.

Each receptive field undergoes filtering using a Gaussian kernel with a standard deviation of $\sigma = 3.0$. When combined with the central feature point position, these filtered receptive fields result in a total of 43 receptive fields forming the descriptor, as illustrated in Figure 6. In this particular application, the distances from each of the 7 rings to the bifurcation point, arranged from the innermost to the outermost, are 4, 6, 8, 13, 18, 26, and 33 pixels, with their respective radii being 1, 2, 3, 4, 6, 9, 13, and 18 pixels. These measurements play a crucial role in determining the spatial arrangement and relative positioning of the receptive fields concerning the bifurcation point within the descriptor's construction.

In initiating the construction of a descriptor using FREAK, the initial step involves calculating the intensities of the receptive fields to ascertain the orientation, denoted as *O*, of the feature point. This orientation determination follows Equation (3):

$$O = \frac{1}{M} \sum_{P_o \in G} \left( I\left(P_o^{r1}\right) - \left(I\left(P_o^{r2}\right)\right) \right) \frac{P_o^{r1} - P_o^{r2}}{\|P_o^{r1} - P_o^{r2}\|} \tag{3}$$

where *G* is the set of all pairs used to compute the local gradient, which are show in Figure 7, *M* is the number of pairs in *G*, and $P_o^{ri}$ is the vector of the spatial coordinate of the center of the receptive field.

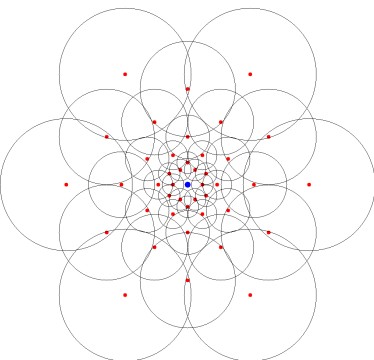

**Figure 6.** Receptive fields utilized to characterize each feature point within fundus images.

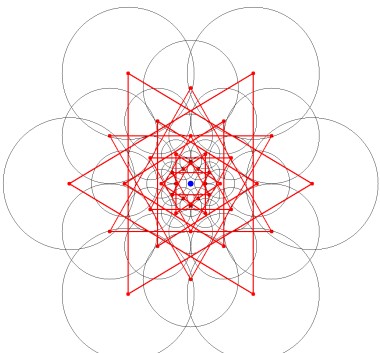

**Figure 7.** Receptive field pairs employed for computing the orientation of feature points within fundus images.

To construct the binary descriptor for each bifurcation, each of the 43 receptive fields is compared with the others, resulting in a 903-bit descriptor. The value of each bit is determined according to Equation (4):

$$T(P_a) = \begin{cases} 1 & I\left(P_a{}^{r1}\right) - I\left(P_a{}^{r2}\right) > 0 \\ 0 & otherwise \end{cases} \tag{4}$$

where the intensities $I\left(P_a{}^{r1}\right)$ and $I\left(P_a{}^{r2}\right)$ correspond to the centers of the smoothed receptive fields within a pair. In prior research [14], it was noted that a 512-bit descriptor provided effective results. Consequently, a selection process is implemented to identify the most relevant pairs. This selection process involves learning, where the determination of the best pairs depends on their correlation, denoted as $\rho$. For this particular application, a correlation threshold of $-0.2 < \rho < 0.2$ is set.

Each feature point is paired with its corresponding nearest neighbor in the other image. In contrast to an alternative approach [9], this proposed method omits the utilization of cross-checking and the Second Nearest Neighbor (SNN) verification techniques. Although cross-checking provided advantages by minimizing false positives and improving match reliability, it runs the risk of discarding potentially valid matches lacking mutual correspondence due to perspective differences, occlusion, or scene discrepancies between fundus images. Similarly, while SNN might also result in the exclusion of valid matches, its effectiveness depends on establishing a reliable threshold between the nearest and second nearest neighbor. Additionally, both these verification methods increase the computational load of the feature matching process. Figure 8 illustrates on the top the resulting matches between two fundus images, illustrating the application of the previously described considerations in the matching process.

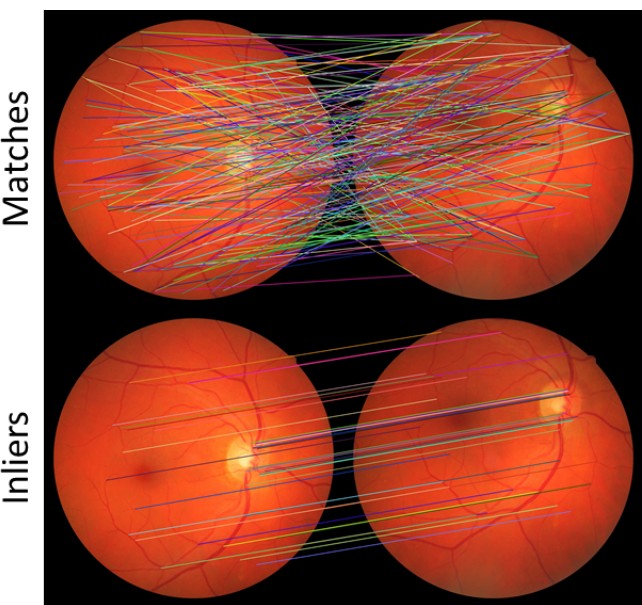

**Figure 8.** Matches acquired for a pair of fundus images using the nearest neighbor approach (**top**) and matches identified as inliers following the application of RANSAC (**bottom**).

### 3.4. Transformation Matrix Computation

The process of determining the geometric relationship between the images involves computing the transformation matrix. Initially, RANSAC distinguishes between inliers and outliers to derive this transformation matrix. Traditionally, RANSAC randomly selects matches to compute a transformation and then determines the number of inliers and outliers for that specific transformation. Yet, in this application, the random selection is restricted, exclusively selecting pairs with a distance greater than a predefined threshold—experimentally set to 20 pixels in this context. This restriction aims to mitigate the influence of localization error on registration error, as outlined in [9]. Figure 8 illustrates on the bottom the resulting inlier matches between the matches shown on the top, demonstrating the application of the previously described considerations in the RANSAC process for separating inliers from outliers.

Some applications may opt for complex transformation models; however, the selection of the best mapping primarily relies on the unique pair of registered images [47]. Hence, considering the advantages associated with a similarity transformation—which includes computational simplicity and requiring minimal correspondences—this model becomes the choice for registering fundus image pairs. This preference is particularly beneficial when faced with challenges inherent in fundus images, such as hidden blood vessels due to disease progression, which restricts the availability of adequate correspondences.

In longitudinal studies and super-resolution imaging involving fundus images, the focus usually centers on aligning images captured from similar or identical perspectives, leading to considerable overlap between the images being registered. Unlike the complexities faced in fundus image mosaicking, where merging multiple images requires meticulous consideration of the retina's curvature and intricate structures, longitudinal studies and super-resolution tasks face less influence from curvature during the registration process.

The considerable overlap found in these scenarios naturally mitigates the impact of the retina's curvature on the alignment process. With significant overlap, the areas of interest within the images display similar perspectives, reducing the impact of the retina's curvature-induced non-linear deformations. Consequently, the need for complex transformation models that precisely address these non-linear distortions diminishes in longitudinal studies and super-resolution imaging. Instead, focus shifts to optimizing computational efficiency, preserving structural integrity, and ensuring consistent and accurate alignment of overlapping regions. This approach simplifies the registration

process, facilitating clearer interpretations and comparative analyses over time or across various image resolutions within these specific applications of fundus imaging.

### 3.5. Image Blending

When aligning fundus images to a unified coordinate system, the blending process aims to create a seamless image, eliminating visible image boundaries. Pixel intensities at corresponding locations across different fundus images are expected to match precisely, but practical scenarios deviate from this ideal state. Despite compensating for variations, certain issues may persist, such as the visibility of fundus image edges due to factors like vignetting, misregistration, and radial distortion. Burt and Adelson's work on multiband blending [48] is a well-established and effective method for creating mosaics, notably reducing blurring and ghosting artifacts.

This multiband blending method, often known as Laplacian pyramid blending, introduces a pyramid-based approach for seamlessly merging images. The approach involves decomposing images into multiple levels or bands of varying spatial frequencies using a Laplacian pyramid. The pyramids consist of numerous layers capturing diverse scales of details, ranging from coarse to fine. The blending process entails combining corresponding layers from these Laplacian pyramids extracted from the fundus images to be merged, from the coarsest to the finest details. By weighting and merging these layers at each pyramid level, the method gradually reconstructs the merged image. This approach utilizes the frequency domain representation of images to smoothly transition low-frequency components from one image to the high-frequency components of another, thereby achieving a seamless and artifact-free blend. Additionally, Laplacian pyramid blending effectively preserves edge details and enhances the visual quality of the resulting merged image. The results derived from the application of this method to registered fundus images are demonstrated in Section 4.5.

## 4. Experimental Results

In this section, an extensive evaluation of the proposed methodology is conducted through experimentation using a well-established public database. Initially, a detailed explanation behind the choice of the specific databases for segmentation and registration evaluation is provided. Subsequently, the evaluation criteria and metrics employed in the analysis of these two tasks are explicated. This is followed by a comprehensive quantitative analysis of the segmentation process, alongside a comparison between the proposed approach for fundus image registration and existing methodologies. Furthermore, notable instances showcasing successful outcomes achieved through the application of the proposed method are presented.

### 4.1. Datasets

The initial step in the fundus registration process involves segmenting blood vessels from the fundus image. This constitutes the basis of feature extraction, achieved through the traditional U-Net architecture trained with the FIVES dataset.

This dataset comprises 800 fundus images, each with a resolution of $2048 \times 2048$ and a field of view of $50°$. Within this set, 200 images are specifically designated for testing purposes. Sourced from 573 patients aged between 4 and 83 years, the dataset encompasses various retinal diseases, including Age-related Macular Degeneration (AMD), Diabetic Retinopathy (DR), and glaucoma, as well as images depicting healthy retinas. The dataset ensures an even distribution of images across each disease category. Notably, approximately 5% of the images were deliberately included due to their poor readability by experienced ophthalmic doctors, aiming to simulate real clinical scenarios. Table 1 offers a comprehensive overview of the FIVES dataset's characteristics, alongside those of other datasets like STARE, DRIVE, ARIA, and CHASEDB1, which were utilized for evaluating the segmentation in the registration method employed in this study. Additionally, Figure 9

presents image examples from each category within the FIVES dataset, along with their corresponding ground-truth blood vessel segmentation masks.

**Table 1.** Summary of publicly available datasets used to evaluate the blood vessel segmentation network for the fundus image registration task.

| Dataset | Year | Number of Images | Resolution | Disease | Annotators |
|---------|------|------------------|------------|---------|------------|
| STARE | 2000 | 20 | $605 \times 700$ | 10 healthy, 10 diseases | 2 |
| DRIVE | 2004 | 40 | $768 \times 584$ | 33 healthy, 7 DR | 3 |
| ARIA | 2006 | 161 | $576 \times 768$ | 61 healthy, 59 DR, 23 AMD | 2 |
| CHASEDB1 | 2011 | 28 | $990 \times 960$ | 28 healthy | 2 |
| FIVES | 2021 | 800 | $2048 \times 2048$ | 200 healthy, 200 AMD, 200 DR, 200 glaucoma | Group |

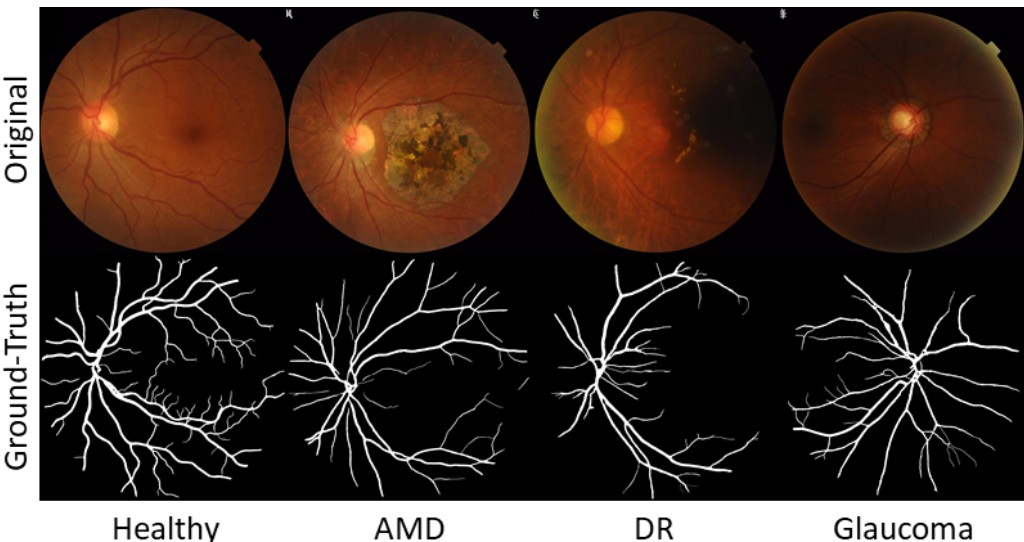

**Figure 9.** Images from the FIVES dataset utilized for both training and testing data during the segmentation phase in the fundus image registration methodology, accompanied by their respective ground-truth annotations.

This analysis exclusively utilizes the Fundus Image Registration (FIRE) dataset for assessment. The choice of this dataset is driven by the limited availability of public databases explicitly designed for fundus image registration. Currently, only four databases contain images suitable for registration (e-ophtha [49], RODREP [50], VARIA [51], and FIRE [52]). Among these datasets, FIRE stands out as the only repository offering ground truth for registration, featuring ten control points. This distinctive attribute enables quantitative evaluation of method performance and facilitates meaningful comparisons with prior research.

The FIRE dataset comprises 129 fundus images, each with a resolution of $2912 \times 2912$ and a field of view spanning 45°, resulting in 134 image pairs. These images originate from 39 patients, ranging in age from 19 to 67 years, distributed across three distinct classes, each serving a specific registration purpose: category S for super resolution, category P for mosaicking, and category A for longitudinal studies. Notably, category A contains registrable pairs reflecting anatomical changes such as vessel tortuosity, microaneurysms, and cotton-wool spots. Table 2 provides an overview of the dataset's characteristics, while Figure 10 visually represents the registrable image pairs, accompanied by their respective ground-truth control points, divided by category.

**Table 2.** Description of each category comprising the FIRE dataset.

|  | Total Image Pairs | Aproximate Overlap | Anatomical Changes |
|---|---|---|---|
| Category S | 71 | >75% | No |
| Category P | 49 | <75% | No |
| Category A | 14 | >75% | Yes |

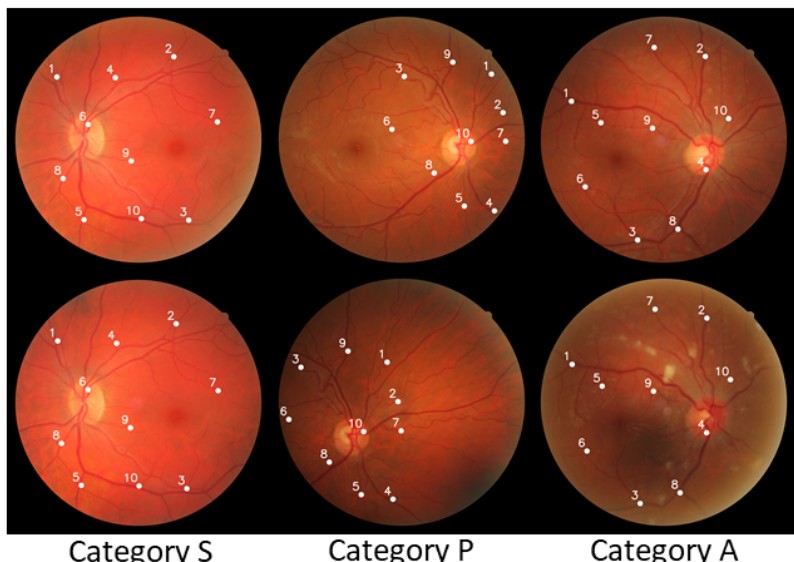

Category S          Category P          Category A

**Figure 10.** Fundus image pairs designated for registration within the FIRE dataset are classified into three distinct categories, each accompanied by its corresponding annotated control points. These control points are established based on the provided coordinates within the dataset, serving as the ground truth.

*4.2. Evaluation Metrics*

The accuracy of blood vessel segmentation is evaluated utilizing the Intersection over Union (IoU) metric, also known as the Jaccard Index. This metric quantifies the extent of overlap between the predicted blood vessel mask and the ground truth mask, as described by Equation (5):

$$\text{IoU} = \frac{\text{predicted} \cap \text{ground truth}}{\text{predicted} \cup \text{ground truth}} \tag{5}$$

where $\cap$ represents the intersection operation, denoting the common region where the predicted blood vessels and ground truth masks overlap, while $\cup$ denotes the union operation, encompassing the entirety of both masks, including their overlapping and non-overlapping areas.

The IoU metric is a fundamental measure for evaluating blood vessel segmentation in fundus images, offering a precise quantitative assessment of their accuracy in delineating blood vessel structures.

In evaluating the extraction of feature points, the uniformity of their spatial distribution is quantified. This assessment employs entropy, a measure reflecting the randomness or disorder within the distribution. Entropy computation involves partitioning the images into fixed-size bins and tallying the points within each bin to establish a probability distribution. Subsequently, this distribution is used to compute the Shannon entropy, as represented in Equation (6):

$$H(X) = - \sum_{x \in X} p(x) \log_2 p(x) \tag{6}$$

where $H$ is the entropy and $p(x)$ is the probability of finding a feature point in bin $x$. In this application, the bin size was set to $32 \times 32$. Concerning the distribution of feature points, higher entropy values signify a more uniform and evenly spread distribution, while lower entropy values suggest a clustered or biased distribution.

Expanding the spatial coverage of feature points in fundus image registration provides multiple advantages. It enhances the robustness of the registration process against deformations, rotations, scaling, or perspective alterations. Additionally, evenly distributed feature points reduce ambiguity in the matching process and are advantageous in situations where fundus image segments may be occluded or altered, such as in longitudinal studies. This dispersion of feature points guarantees the availability of reference points for matching, even in affected or obscured areas of the image.

On the other hand, the fundus registration evaluation method follows the guidelines outlined in [17], leveraging strategically positioned control points identified by experts across overlapping fundus image pairs. Registration error is computed as the average distance between each control point in the target image and its corresponding point in the source image after registration. This assessment was conducted across the entire dataset and separately for different categories.

To illustrate accuracy across different error thresholds, a 2D plot was created. Here, the x-axis indicates the error thresholds, while the y-axis shows the percentage of image pairs successfully registered at each threshold. Successful registration occurs when the error is below the defined threshold. The resulting curve displays the success rate concerning the desired accuracy, enabling comparison between methods and assisting in selecting the most appropriate approach based on specific accuracy needs. Additionally, the curve offers a thorough evaluation by assessing the area under the curve (AUC).

*4.3. Segmentation Performance*

Accurate segmentation of blood vessels in fundus images is crucial for facilitating precise image registration. This subsection presents an evaluation of the blood vessel segmentation performance achieved through the proposed methodology. Leveraging a robust segmentation framework that integrates U-Net architecture, this approach aims to delineate blood vessel structures accurately. Through comprehensive evaluation using diverse metrics, including visual assessments and quantitative evaluation, a detailed examination of the segmentation results is provided.

The U-Net model for blood vessel segmentation underwent several training iterations using various datasets and adjustments. Initially, 1000 randomly chosen patches from the untreated green channel were trained for 20 epochs, resulting in an IoU of 0.5152. To improve performance, the training was extended to 30 epochs, leading to an IoU increase to 0.5436, with loss and accuracy scores of 0.1078 and 0.9670, respectively, along with comparable validation results.

To refine the segmentation quality, CLAHE was applied to the same 1000 patches, resulting in an average IoU of 0.6165 after 20 epochs. Subsequently, 2000 preprocessed patches, with an emphasis on bifurcation areas, were utilized, achieving an IoU of 0.7178. However, this approach displayed slightly reduced training and validation scores for loss and accuracy. This experiment demonstrated improved IoU by concentrating on more extensive blood vessel areas during training. Figure 11 depicts the accuracy and loss curves for this experiment, which are crucial metrics for evaluating the model's performance. These curves illustrate the initial training phase before the fine-tuning phases.

Employing the weights from the most successful experiment, a fine-tuning stage was initiated. The first fine-tuning utilized 2000 patches focusing on bifurcation areas, integrating Gamma correction and CLAHE, resulting in an IoU of 0.7491 over 20 epochs, demonstrating improved scores across all metrics. Subsequently, the second fine-tuning, featuring a reduced learning rate and additional patches, achieved an IoU of 0.7559 after 5 epochs, maintaining competitive loss and accuracy values. Notably, this final fine-tuning, limited to 5 epochs, did not yield a significant increase in IoU beyond this point.

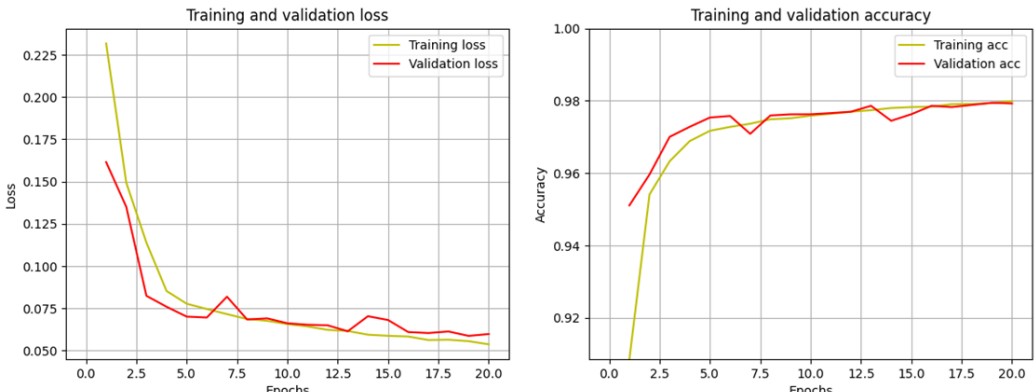

**Figure 11.** The loss curve and accuracy curve generated during the initial phase of training for the U-Net model in the blood vessels segmentation process within the fundus image registration method.

Figure 12 showcases sample qualitative outcomes of the segmentation phase, complemented by Table 3, which outlines the IoU results across various datasets utilized for evaluating this segmentation network.

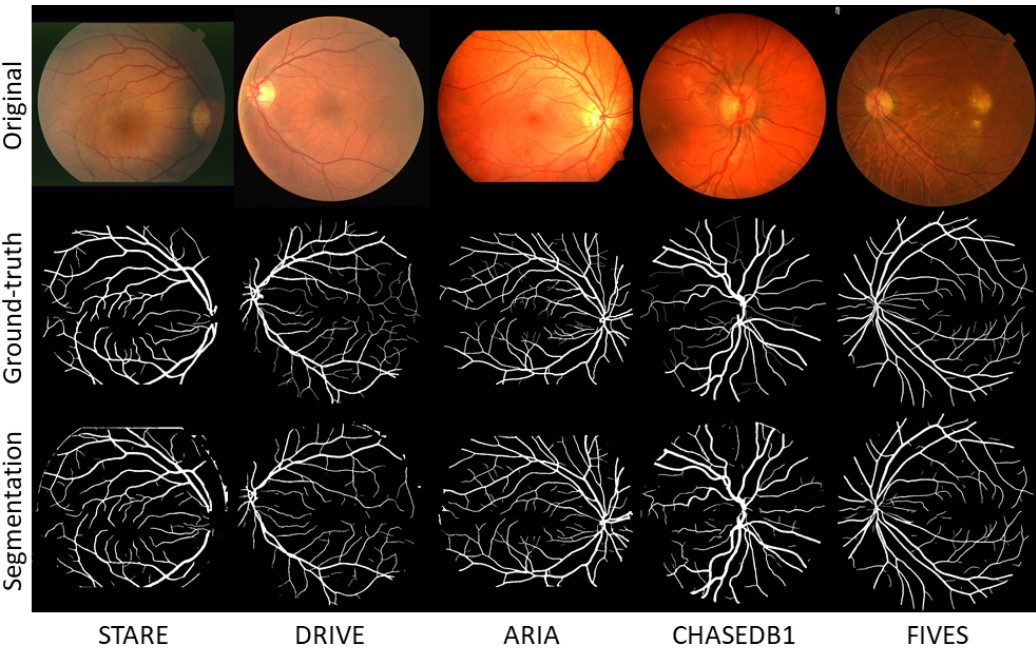

**Figure 12.** Example results for the segmentation with U-Net employed as part of the fundus image registration method.

**Table 3.** Datasets utilized to evaluate the performance of the U-Net architecture in blood vessel segmentation.

| Dataset | Number of Images | Intersection over Union |
|---------|------------------|-------------------------|
| STARE | 20 | 0.5402 |
| DRIVE | 40 | 0.5602 |
| ARIA | 161 | 0.4532 |
| CHASEDB1 | 28 | 0.6326 |
| FIVES | 200 | 0.7559 |

The disparity in performance between the U-Net model trained on the FIVES dataset and its applicability to established fundus blood vessel segmentation datasets highlights the intricate challenges associated with model adaptability and generalization in medical

image analysis. While the model excels within the controlled parameters of the FIVES dataset—characterized by unique imaging conditions and diverse pathology presentations—its translation to other datasets reveals intricate challenges. Variations in dataset characteristics, including differences in image quality, diverse pathologies, and demographic representations, present challenges for seamless model generalization. Additionally, differences or inconsistencies in annotation quality and precision across evaluation datasets pose significant obstacles, potentially impacting the model's adaptability. This issue is highlighted in [16], where improper labeling from the DRIVE dataset is noted. Furthermore, the U-Net model's effectiveness within the FIVES dataset may arise from a degree of overfitting, in which it has adjusted to the specific unique features of that dataset during training, resulting in reduced adaptability to the distinct characteristics presented by evaluation datasets. These challenges underscore the urgent need for more comprehensive, diverse, and meticulously annotated datasets, representative of real-world variability, to foster the development of robust and adaptable segmentation models in fundus image analysis. This approach can help mitigate issues related to dataset bias, annotation quality, domain shift, and overfitting.

Quantitative evaluation of blood vessel segmentation in the FIRE dataset is not possible because there is no ground truth data available for vessel segmentation. However, Figure 13 displays examples from every category of fundus images in the FIRE dataset, showing their respective segmentations produced using the proposed method.

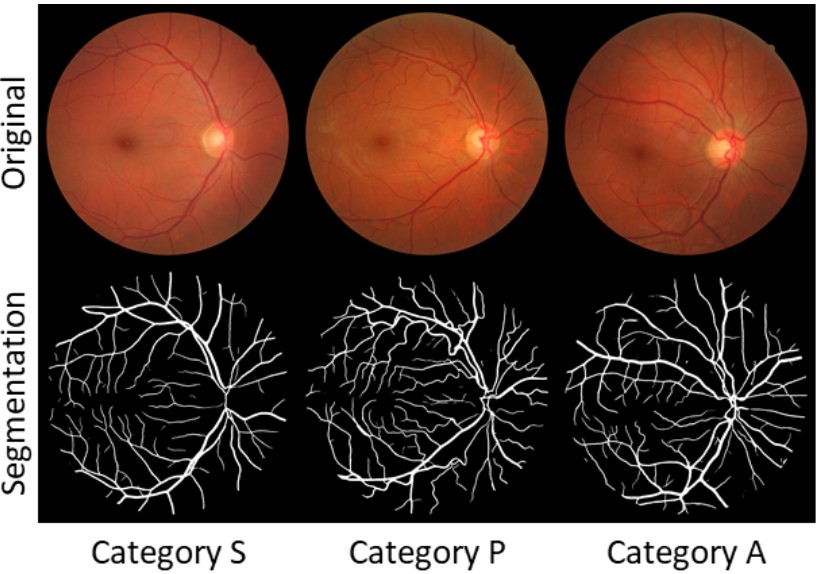

**Figure 13.** Example results for the segmentation with U-Net for different categories of FIRE dataset.

*4.4. Feature Extraction and Feature Description*

The evaluation of feature extraction involves analyzing the spatial distribution across fundus images, comparing methods such as SIFT [18], Oriented FAST and Rotated BRIEF (ORB) [53], and SBP-FIR [9]. Table 4 outlines the average detection of feature points within the FIRE dataset's fundus images for these methods and the proposed approach, along with their respective mean entropy values.

**Table 4.** Comparison of mean feature point count and entropy values among ORB, SIFT, SBP-FIR, and the proposed method.

|  | ORB | SIFT | SBP-FIR | Proposed Method |
|---|---|---|---|---|
| Average number of feature points | 241 | 164 | 125 | 222 |
| Average entropy | 5.6809 | 5.9600 | 6.4245 | 7.0137 |

The findings from Table 4 reveal that the proposed method achieves a balance in feature extraction. It consistently detects more features on average than SIFT but fewer than ORB, yet it achieves a superior distribution across fundus images. In image registration, an abundance of feature points poses disadvantages, including increased memory usage and processing time due to the need for describing and matching more features, thus escalating computational complexity. Conversely, a scarcity of feature points limits method coverage and robustness, potentially compromising the capture of sufficient information to tolerate deformations or viewpoint changes. Moreover, a reduced number of feature points could amplify the impact of even a small number of incorrect correspondences on alignment accuracy. Therefore, the aim in registering fundus images is to maintain a balance, as evidenced by comparing this method with other state-of-the-art approaches like SIFT and ORB. An example of feature extraction for each method from Table 4 is illustrated in Figure 14.

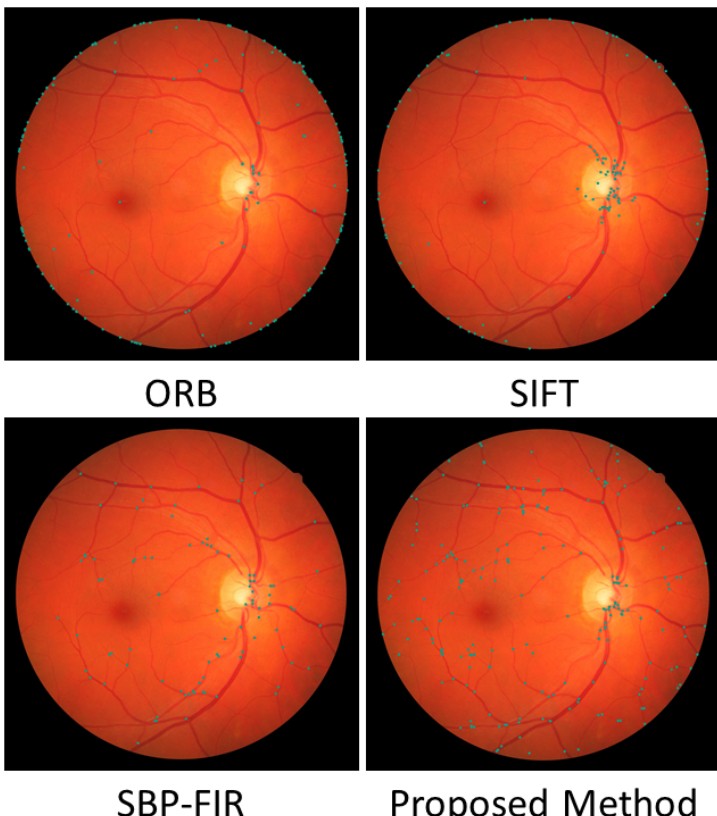

**Figure 14.** Examples showcasing feature extraction methodologies corresponding to each method detailed in Table 4.

Finally, the FREAK feature descriptor method demonstrates the least correlated receptive fields, an essential aspect in reducing the descriptor to 512 bits. Figure 15 displays the paired receptive fields used to construct the descriptor across at least 50% of the fundus image pairs in the FIRE dataset.

The observation from Figure 15 reveals a clear vertical pattern in the pairs employed to construct the descriptor. This trend arises from the rotational adjustment made during the angle computation of the points, demonstrating the FREAK descriptor's capacity to maintain invariance despite pattern rotations.

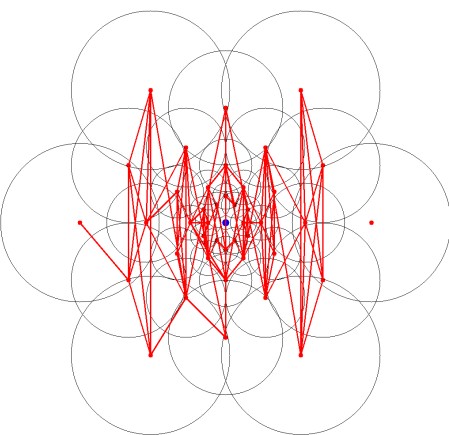

**Figure 15.** Receptive field pairs employed during the feature description stage of the FREAK descriptor, illustrating those utilized in at least 50% of the feature points across the entire FIRE dataset.

*4.5. Registration Performance on the FIRE Dataset*

This section explores the assessment of registration accuracy, a fundamental criterion in assessing the effectiveness of the proposed fundus image registration method. The registration error is a crucial metric that reflects the alignment quality between registered images, serving as an essential measure of the method's precision and reliability. This comprehensive analysis examines the registration success across different thresholds, revealing the method's robustness under varied alignment conditions and specifications. Supported by visual representations displaying registered images from diverse categories within the FIRE dataset, this section provides an evaluation providing valuable insights into the method's performance across a range of alignment scenarios.

Figure 16 displays the performance of various registration methods on the FIRE dataset, assessing accuracy for each category and overall. Additionally, Table 5 presents the area under the curve (AUC) for all compared methods, along with details on execution time and the transformation model used in each approach.

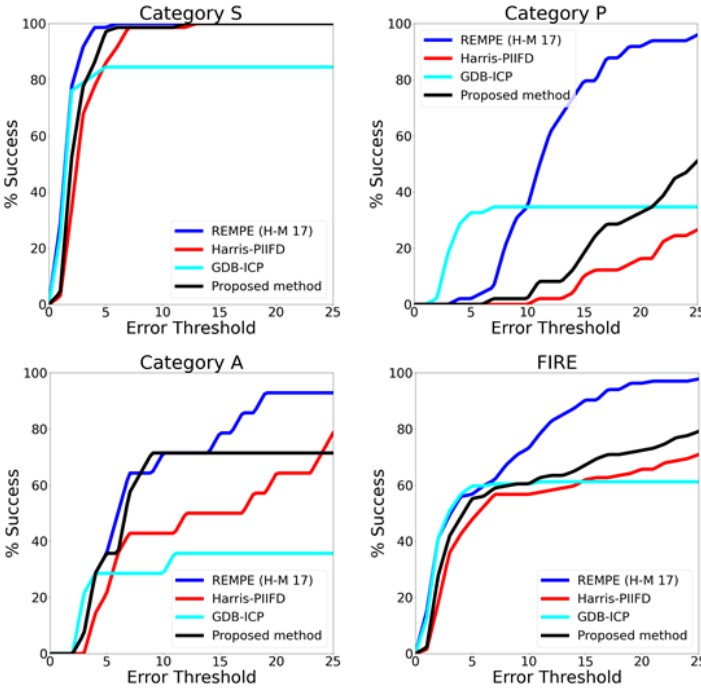

**Figure 16.** Evaluation of retinal image registration techniques on the FIRE dataset, encompassing the outcomes of the proposed approach.

**Table 5.** Comparison of the Area Under the Curve (AUC) values for diverse fundus image registration methods, along with their corresponding execution times in seconds and the respective transformation models utilized.

| | Category S | Category P | Category A | FIRE | Execution Time | Transformation Model |
|---|---|---|---|---|---|---|
| REMPE (H-M 17) [20] | 0.958 | 0.542 | 0.660 | 0.773 | 198 | Ellipsoid eye model |
| Harris-PIIFD [54] | 0.900 | 0.090 | 0.443 | 0.553 | 13 | Polynomial |
| GDB-ICP [55] | 0.814 | 0.303 | 0.303 | 0.576 | 19 | Quadratic |
| ED-DB-ICP [56] | 0.604 | 0.441 | 0.497 | 0.553 | 44 | Affine |
| SURF+WGTM [57] | 0.835 | 0.061 | 0.069 | 0.472 | – | Quadratic |
| RIR-BS [58] | 0.772 | 0.004 | 0.124 | 0.440 | – | Projective |
| EyeSLAM [59] | 0.308 | 0.224 | 0.269 | 0.273 | 7 | Rigid |
| ATS-RGM [60] | 0.369 | 0.000 | 0.147 | 0.211 | – | Elastic |
| SBP-FIR [9] | 0.835 | 0.127 | 0.360 | 0.526 | – | Similarity |
| Proposed Method | 0.903 | 0.159 | 0.562 | 0.596 | 96 | Similarity |

Yang et al., as detailed in [55], developed a technique aimed at registering diverse scenes, from natural landscapes and built environments to medical imagery such as fundus images. Their method relies on feature points such as corners. Depending on the particular image pair being examined, they utilize different transformation models, including similarity, affine, homography, and quadratic models.

Similarly, the study by Chen et al. [54] utilizes corners identified through a Harris detector as feature points for registration. Their approach adjusts different models depending on the number of matches obtained from the feature extraction, description, and matching processes. Their descriptor emphasizes the primary orientation of the points and relies on gradients.

Other comparative studies, such as [56], are inspired by [55], refining the generation of keypoint matches during initialization. This adjustment involves extracting Lowe keypoints from the gradient magnitude image and enhancing the keypoint descriptor by integrating global-shape context through edge points. The purpose is to overcome a limitation in [55], where its performance faces challenges when dealing with image pairs showing substantial non-linear intensity differences.

In [58], a novel approach to fundus image registration is presented, focusing on a unique structural feature. Unlike traditional methods that depend on single bifurcation point angles, this approach employs a structure-matching technique. It employs a master bifurcation point and its three connected neighbors to create a distinct vector, consisting of a normalized branching angle and length. This vector remains stable under typical transformations, minimizing uncertainty in matching and helping to resolve ambiguous matching situations. Its simplicity and efficiency enable autonomous implementation or integration with other methods, providing versatility in hybrid or hierarchical setups.

Figure 16 shows significant differences in results among different categories. Categories S and A, characterized by extensive overlapping regions, naturally produce more correspondences, resulting in improved matching performance. In contrast, category P presents challenges due to the limited potential matches within the overlapping area of fundus images. Likewise, category A encounters potential reductions in matches due to morphological changes observed in longitudinal studies.

In category S, the GDB-ICP method demonstrates exceptional performance, achieving numerous successful registrations with minimal errors. However, it achieves successful registration for only 84.5% of the category. In contrast, the proposed method, along with Harris-PIIFD and REMPE, achieves a 100% success rate.

In category A, REMPE emerges as the leader, boasting a success rate of 92.85%. It is followed by Harris-PIIFD and the proposed method, achieving 78.57% and 71.42%, respectively. However, the proposed method outperforms both alternatives in scenarios with lower thresholds.

In category P, REMPE surpasses other methods, except for GDB-ICP, which outperforms REMPE in registration error thresholds less than 10 pixels. The proposed method achieves a 51.02% success rate in category P, while GDB-ICP and REMPE achieve 34.69% and 95.91%, respectively.

Table 5 provides a comprehensive examination of the registration performance of these methods, incorporating the AUC metrics alongside details about the transformation model and execution time. It is worth noting that despite REMPE's high success rate in registering fundus images, it exhibits considerably slower execution times compared to the other methods discussed. Our proposed method, while achieving comparable performance with REMPE in certain categories, significantly reduces the execution time by over 50%, improving efficiency without compromising efficacy. Figure 17 illustrates sample results for each category within the FIRE dataset, showcasing matches and the results after registration.

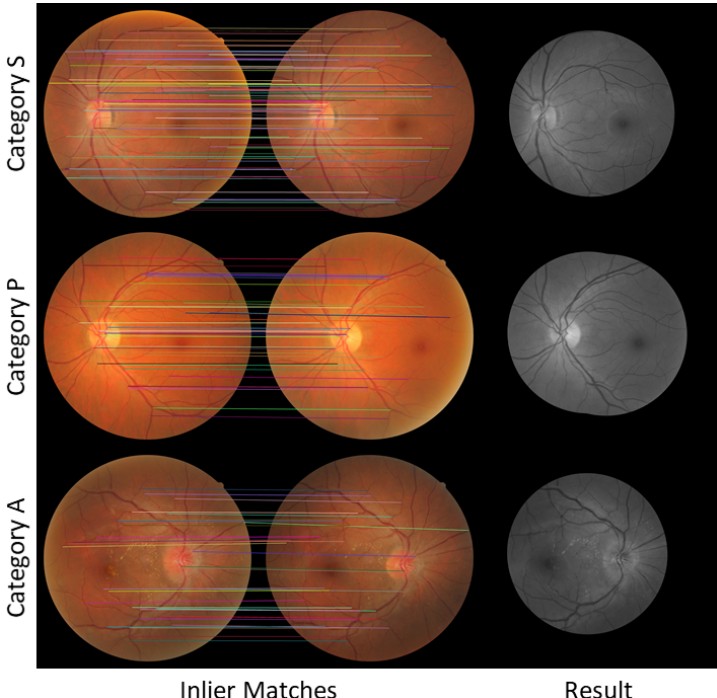

**Figure 17.** Image pairs within the FIRE dataset showcase registration outcomes for super-resolution imaging (**top row**), image mosaicking (**central row**), and longitudinal study (**bottom row**).

## 5. Discussion

This study aimed to create a novel fundus image registration method, leveraging bifurcations as feature points alongside a deep learning-based approach for blood vessel segmentation. The main objective was to establish a geometric relationship between fundus images by extracting features to aid in their alignment. Furthermore, the effectiveness of FREAK as a keypoint descriptor in the registration process was investigated.

The investigation produced significant insights. Particularly, the binary descriptor FREAK proved highly effective in fundus images, surpassing expectations given the texture limitations. Moreover, a clear correlation emerged between the quality of blood vessel segmentation, facilitated by U-Net, and the subsequent registration accuracy. This relationship was underscored by a qualitative comparison with previous methodologies [9,10].

This aligns with the existing literature, demonstrating that bifurcations acquired from blood vessel segmentation using a similarity transformation can compete with approaches employing more complex feature extraction and models. These results imply that even simpler methods can achieve competitive outcomes in fundus image registration.

The practical implications of this research are significant. The decrease in the execution time of fundus image registration moves it closer to integration into clinical practice. Additionally, the comprehensive evaluation conducted in this study establishes a benchmark for future methodologies, assisting in assessing their suitability for clinical adoption.

However, a notable limitation of this study is the use of separate datasets for blood vessel segmentation and registration evaluation. This setup makes it challenging to establish a direct correlation between enhanced segmentation and improved registration accuracy. Future research efforts should prioritize creating and utilizing datasets that include both blood vessel segmentation ground truth and control points, enabling integrated evaluations.

In terms of future directions, investigating the possibility of automatically choosing transformation models according to fundus image attributes presents an exciting prospect. Since the most suitable transformation model might differ depending on the characteristics of fundus images, creating adaptable models could greatly improve registration accuracy across various fundus image types.

In conclusion, this study highlights the potential of simplified approaches in achieving effective fundus image registration and suggests possible research directions for refining registration techniques suitable for practical clinical implementation.

## 6. Conclusions and Future Work

This study reveals the potential of simplified yet robust fundus image registration techniques, utilizing bifurcations derived from blood vessel segmentation. The substantial progress in reducing execution time represents a significant step towards the practical integration of these methods into clinical workflows. However, a critical limitation arises from the use of different datasets for segmentation and registration evaluation, hindering a complete demonstration of the causal relationship between improved segmentation and registration accuracy. Future investigations require unified datasets containing both ground truth segmentation and registration control points to enable a more comprehensive validation of these techniques.

Research in this field necessitates attention on multiple fronts. Firstly, refining registration algorithms to achieve an optimal balance between reduced execution times and sustained accuracy is crucial. This effort would enable smooth integration of these methodologies in clinical practice. Secondly, defining precise accuracy benchmarks is a critical task. Establishing these benchmarks is essential not only for validation but also for the integration of these methodologies into clinical workflows. The absence of defined accuracy standards poses a significant challenge in determining the suitability of these methods for real-world clinical applications. Hence, future research must prioritize the establishment of these accuracy benchmarks to optimize the validation and adoption of these techniques in clinical practice.

**Author Contributions:** Conceptualization, J.E.O.-A., W.D. and Y.P.; Methodology, J.E.O.-A. and Y.P.; Software, J.E.O.-A.; Validation, J.E.O.-A., L.W. and W.D.; Formal analysis, J.E.O.-A., L.W., W.D. and Y.P.; Data curation, L.W.; Writing—original draft, J.E.O.-A.; Writing—review & editing, W.D. and Y.P.; Visualization, W.D.; Supervision, W.D. All authors have read and agreed to the published version of the manuscript.

**Funding:** This research received no external funding.

**Data Availability Statement:** Data are contained within the article.

**Conflicts of Interest:** The authors declare no conflicts of interest.

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
