# Peer review of "Enhanced Vascular Bifurcations Mapping: Refining Fundus Image Registration"

_electronics, doi:10.3390/electronics13091736_

Round 1
Reviewer 1 Report
Comments and Suggestions for Authors
This paper proposes a novel fundus image registration algorithm. The superiority of the proposed algorithm is verified by comparative experiments on the Fundus Image Registration Datasets. However, to improve the quality of this paper in the future, the following suggestions should be taken into account

Moderate editing of English language required
Author Response
Please find the correspondence attached.

Reviewer 2 Report
Comments and Suggestions for Authors
The manuscript “Enhanced Vascular Bifurcations Mapping: Refining Fundus Image Registration” by Ochoa-Astorga et al. is devoted to developing a novel image registration method for fundus imaging.
The work is interesting in general as it combines multiple methods. However, it is far from being ready for publishing in its current form. It needs to be rewritten almost entirely (see comment 2).
1. The novelty of the work is unclear. For example, multiple works that use U-net architecture are cited. The contribution and novelty need to be explicitly stated in the Introduction.
2. While the work is generally comprehendible, it is very difficult to read. The language and sentence construction are very cumbersome and difficult to follow. Here are just a few examples:
a) The paper contains multiple repetitive themes, even within a single section. For example, in lines 62-65, “In contrast to prior methodologies that used thresholding-based segmentation and Frangi filter [9–11] as the foundation for blood vessel delineation in fundus images, this approach initiates with the utilization of U-Net [12] to identify the specific region of interest within the image.”. The same idea is repeated in lines 90-92: ‘Departing from traditional approaches relying on thresholding-based segmentation or Frangi filter applications, this method introduces a paradigm by initiating the region-of-interest identification through U-Net [12].’
b) Many passages are absolutely unclear. For example, Lines 120-122 “This work was evaluated on the Fundus Image Registration Dataset (FIRE) dataset [17], which was also employed in [20], being the latter one of the top leaders in registration on this dataset” is almost nonsensical. Which work? What are “top leaders in registration”?
c) Lines 126-128: “While commendable for its precision, a discernible drawback emerges in its approach to feature detection, employing dual feature detectors, thereby augmenting the amount of feature points.” What does it mean?
d) Line 232: Why is the method revised? Did you mean “proposed”?
e) Line 268-270: “… many times are better not only regarding time [44] but also in accuracy, in which deep learning approaches are even better than human experts in retinal vessel segmentation.” Many time are better??? Did you mean faster? “many times are better… in accuracy…”??? How is it possible?
Some minor shortcomings
3. Why is the pre-processing step missing from Fig 1 and its description?
4. What do “1” and “-1” stand for in Fig 5? It needs to be clearly explained in the text.
5. Line 537: Why “To enhance the segmentation quality, CLAHE was applied to the same 1000 patches”? Based on the method description, CLAHE should be universally applied to all images during the pre-processing step.
Comments on the Quality of English LanguageThe language is cumbersome
Author Response

(The authors gave the same response as above.)

Round 2
Reviewer 1 Report
Comments and Suggestions for Authors
Accept in present form.
Comments on the Quality of English LanguageMinor editing of English language required
Author Response
English errors have been modified.
Reviewer 2 Report
Comments and Suggestions for Authors
The article has been improved in this round of revisions. While there are certain doubts about the proposed methods (e.g., noticeably worse performance on other datasets- see Table 3), in general, the article can be published if several minor deficiencies are addressed:
1. Reference 3 contains some dots in the authors' list. The common approach is to use "et al." for long authors lists.
2. References 3, 26, 39, 42 are incomplete.
3. The format of references is inconsistent. For example, Ref. 33-35 use a completely different format
Comments on the Quality of English LanguageEnglish is fine
Author Response
The response is attached to this email
